# Copper Increases the Cooperative Gating of Rat P2X2a Receptor Channels

**DOI:** 10.3390/ph17121590

**Published:** 2024-11-26

**Authors:** Elias Leiva-Salcedo, Denise Riquelme, Juan Pablo Huidobro-Toro, Claudio Coddou

**Affiliations:** 1Department of Biology, Faculty of Chemistry and Biology, Universidad de Santiago de Chile, Santiago 9170022, Chile; elias.leiva@usach.cl (E.L.-S.); denise.riquelme@usach.cl (D.R.); 2Departamento de Ciencias Biomédicas, Facultad de Medicina, Universidad Católica del Norte, Coquimbo 1781421, Chile; 3Millennium Nucleus for the Study of Pain (MiNuSPain), Santiago 8331150, Chile

**Keywords:** cooperativity, P2X2, P2X2a, single channel, copper, trace metals

## Abstract

**Background/Objectives:** P2X receptor channels are widely expressed in the CNS, where they have multiple functions in health and disease. The rat P2X2a (rP2X2a) receptor channel is modulated by copper, an essential trace element that plays important roles in synaptic modulation and neurodegenerative disorders. Although essential extracellular amino acids that coordinate copper have been identified, the exact mechanism of copper-induced modulation has not been yet elucidated. **Methods**: We used HEK293T cells expressing rP2X2a channel(s) and performed outside-out single-channel and whole-cell recordings to explore copper’s effects on rP2X2 currents and determine whether this metal can increase the cooperative gating of rP2X2a channel. **Results**: In whole-cell recordings and in patches containing 2 or 3 rP2X2a channels, copper enhanced the ATP-induced currents, significantly reducing the ATP EC_50_ and increasing the Hill coefficient. Moreover, copper increased the apparent P_o_ in patches containing two or three channels. By contrast, in patches containing only one rP2X2a channel, we did not observe any significant changes in ATP EC_50_, the Hill coefficient, or P_o_. **Conclusions**: Copper modulates the gating of rP2X2a channels, enhancing interchannel cooperativity without altering single-channel conductance or P_o_. This novel regulatory mechanism could be relevant for understanding the role of P2X2 channels in physiological and pathological processes.

## 1. Introduction

More than 50 years after the proposal of Purinergic Signaling by Geoffrey Burnstock [1], the study of the different components, including purinergic receptors from P2X and P2Y families [2], still attracts the attention of researchers because of their involvement in several physiological and pathological events, including synaptic transmission, vascular physiology, gastrointestinal physiology, reproduction, cancer, pain, cardiovascular diseases, respiratory diseases, and neurodegenerative diseases, among others [2,3,4,5].

Besides its catalytic and regulatory functions, copper plays important roles in synaptic transmission, regulating the activity of ion channels, and it is also involved in the pathogenesis of neurodegenerative disorders such as Alzheimer’s and prion diseases. Copper is stored and released in synaptic vesicles, reaching concentrations up to 100–300 µM in the synaptic cleft [6,7,8].

P2X receptors are trimeric ligand-gated, non-selective cation channels activated by extracellular ATP and permeable to Na^+^, K^+^ and Ca^2+^. Despite the high structural homology between P2X receptors, they show important differences in terms of ATP affinity, desensitization properties, and the effects exerted by divalent metals, such copper and zinc, which act as positive or negative allosteric modulators on these receptor channels, with each P2X subtype displaying a unique modulation profile [2,9]. These metals act at P2X receptor channels binding at specific allosteric sites, most of them located in the receptor extracellular domain [10].

In rats, the P2X2 receptor channel exists in two splice variants, P2X2a and P2X2b, which differ in their desensitization properties. P2X2b, the shorter variant, lacks a segment of 69 amino acids within the intracellular C-terminal domain, leading to a faster desensitization compared with the full-length P2X2a [2,11,12]. Despite this difference in desensitization, both variants retain similar ATP potency and sensitivity to modulation by divalent metals such as zinc and copper [2]. The rP2X2a receptor channel is a slow desensitizing channel, with an ATP EC_50_ of 3–10 µM [10]. Both zinc and copper potentiate the ATP-induced currents on this receptor, and the allosteric binding site for these metals has been identified and includes two extracellular histidines, H192 and H319, from two neighboring subunits that coordinate zinc or copper; this interaction stabilizes ATP binding to its orthosteric site, resulting in current potentiation [13,14,15]. The rP2X2a receptor is the only member of the P2X family that is potentiated by copper; other P2X subtypes such as rP2X4 or rP2X7 are instead inhibited by this divalent metal [9]. The ligand-gated rP2X2a receptor is widely expressed in the central and peripheral nervous system [16] and participates in the modulation of long-term potentiation [17], synaptic integration, and the spontaneous release of synaptic vesicles [18]. Non-controlled activation of rP2X2a leads to neuronal death after brain injury [19], thus suggesting that the modulation of rP2X2a is critical for neurotransmission and normal brain function [20].

At the molecular level, it has been shown that rP2X2a activates cooperatively after ATP binding, which results in the non-linear increase in its current; the number of channels interacting between them increases the kinetics of the active channels and favors the open state of the channel. This suggests that the interaction between rP2X2a is critical for the current gain [21,22]. Although P2X2 is the only purinergic receptor channel described to exhibit cooperative gating, this phenomenon is an important regulatory mechanism for ligand- and voltage-gated ion channels. Positive coupling, the most common form, has been observed in ligand-gated channels (e.g., nAChRs, AMPA receptors, and RyRs) [23,24,25], among others, while negative coupling is rare and has been observed in Na^+^ channels [26]. This mechanism of gating may have important implications in pathology, as cooperative gating could increase neuronal depolarization or increase intracellular Ca^2+^, inducing neuronal death. However, the effects of this cooperative gating in the presence of positive allosteric modulators, such as copper, have not been yet established.

Copper is involved in memory impairment [27], Alzheimer’s disease [28,29], and neuronal death [30], suggesting that the combined response of ATP and copper over rP2X2a could be involved in this process. To address the mechanism by which copper increases rP2X2a currents, we used single channel recordings in HEK 293T cells overexpressing rat rP2X2a receptors, and we showed that copper increases the open probability by expanding the mean open channel time without changing the conductance. Moreover, copper increases the cooperative gating, thus augmenting the rP2X2a current. Our results suggest a key role of copper in the rP2X2a cooperative gating, determining the potentiation of the macroscopic current observed after its activation.

## 2. Results

### 2.1. Copper Potentiates ATP-Induced rP2X2a Currents

First, we determined the effect of copper on ATP-evoked rP2X2a currents. We performed an ATP concentration-response curve in the presence of 10 µM copper and measured the current using a patch clamp whole-cell configuration. We found a robust ATP response (Figure 1A) that was concentration-dependent and is potentiated by copper (EC_50_ ATP = 5.5 ± 0.4 µM, n = 5; EC_50_ ATP-copper = 1.6 ± 0.07 µM, n = 5; *p* < 0.05, Figure 1B). Furthermore, we found an increase in the Hill number (ATP = 1.6, ATP-copper = 2.2, Figure 1B), suggesting an increase in cooperativity [22].

### 2.2. Copper’s Effect on rP2X2a at Single-Channel Level

Changes in whole-cell currents depend on the number of channels, the unitary current, and the open probability (P_o_) of the channel; thus, we first investigated the effect of 10 µM copper on rP2X2a unitary currents by performing outside-out patch clamp in HEK293T cells expressing the rP2X2a receptor channel. In the isolated patches, we observed that 10 µM ATP activates an inward current of 2.4 pA that is completely reversed after washout. Conversely, 10 µM copper does not activate any current, but 10 µM ATP + 10 µM copper activates a similar current as ATP but with a higher number of transitions between open and closed states (Figure 2A). The cord conductance was not changed by 10 µM copper at any of the holding potentials tested (Figure 2B), suggesting that copper’s effect on ATP-gated rP2X2a currents is not related to changes in conductance.

### 2.3. Effect of Copper in ATP-Induced Single-Channel Current of rP2X2a

To further explore the mechanism of copper potentiation, we performed ATP concentration-response curves (1–300 µM ATP) in the presence of 10 µM copper on outside-out patches containing a single rP2X2a channel. We considered a single channel-containing patch when only one level of current transition occurred at 300 µM ATP. We found that the maximal P_o_ was ~0.6, reached in both cases at 100 µM ATP (Figure 3A,B) [21]. Next, we observed that copper had no effect on the sensitivity for ATP (EC_50_ ATP = 10.6 ± 3.1 µM, n = 5; EC_50_ ATP-copper = 5.2 ± 1 µM, n = 5, *p* < 0.05, Figure 3C) or on the Hill number (nH ATP = 0.7, n = 5; nH ATP-copper = 1, n = 5, *p* < 0.05, Figure 3C). Furthermore, we found that 10 µM copper did not change the conductance at any of the ATP concentrations tested (Figure 3D). Together, these results show that at the single-channel level, copper has no effect on rP2X2a, suggesting that other parameters are involved in copper potentiation observed in the whole-cell recordings.

### 2.4. Copper Increases Cooperative Gating in rP2X2a Receptor

A previous report indicated a non-independent gating of rP2X2a, which is manifested when more than one channel is present in the outside-out patches [22]. Therefore, we hypothesized that copper might increase the cooperative gating, thus resulting in the current potentiation observed at the whole-cell level. To assess this hypothesis, we isolated outside-out patches with two or three channels and evaluated them for non-independent gating. The most frequent recordings were those expressing three channels. We observed lower current overlap at low ATP concentrations, but higher concentrations increased the number of events that overlapped (Figure 4A,B). The application of copper increased ATP sensitivity and increased the Hill number (EC_50_ ATP = 12.1 ± 2.4 µM, nH = 1.2; EC_50_ ATP-copper = 1.3 ± 0.2 µM, nH = 2.1, n = 4, *p* < 0.006, Figure 4C). Moreover, we found that the maximal P_o_ reached ~0.7, a higher magnitude than a single channel. This increase in the Hill coefficient suggests that copper increased cooperative gating, explaining the left shift in the ATP concentration-response curve. Additionally, the analysis of the mean open time at a single channel containing patches showed no difference between ATP (10 µM) with or without 10 µM copper, but in multiple channels containing patches, the mean open time was higher at all levels of current (Figure 4D).

To determine the effect of copper in the cooperative gating of rP2X2a receptors, we determined the independence of the channel gating in patches containing multiple channels, by performing a binomial analysis [22] to compare the expected independent gating (binomial) and the experimental P_o_; thus any deviation from 1 means that the experimental P_o_ is non-independent, regardless of the number of the ratio (lower or higher than 1). We found that the ratio P(k)_exp_/P(k)_binomial_ increased, with an increase in channel number (k) in the presence of 10 µM copper, showing a deviation from the binomial distribution and suggesting non-independent gating of the channel (Figure 5A).

To quantify the magnitude of the cooperativity elicited by copper, we used the Ising model for cooperative behavior [31]; this model provides a framework for studying systems of protomers in which the nearest neighbors’ elements interact with each other. We plotted multiple channel current variance as a function of P_o_; at P_0.5_, the current variance and thus the energy of the system was maximal; using the one-dimensional Ising model, we obtained the parameters of interacting energy (ϵ_BB_). We found that ϵ_BB_ in multiple channels was −0.4 kT, showing positive cooperativity, as expected [22]. Moreover, the ϵ_BB_ for ATP-copper was −1.4 kT, showing higher positive cooperativity when multiple channels were present in the patch (Figure 5B). Altogether, this analysis shows that copper increased the interaction energy of ATP-induced current 3.5-fold, suggesting an increased cooperative gating.

## 3. Discussion

Our results demonstrate that copper enhances ATP-induced rat rP2X2a currents by increasing its cooperative gating. This potentiation is evident only when multiple channels are present in the membrane, suggesting that cooperative gating is the primary mechanism underlying the copper effect. This interaction likely occurs between different rP2X2a channels rather than within individual channels, as copper-induced potentiation was found in whole-cell recordings and in patches containing two or three rP2X2a receptor channels, but not when only one channel was present.

Earlier studies have shown that ATP increases the non-independent gating of rP2X2a channels when multiple channels are present in the patch [22]. Additionally, a recent study showed that this cooperative gating depends on the subunit conformational flip to the transmembrane gate reaction, involving Histidine 319 [32]; however, this has not been probed at the single-channel level. Consistent with these findings, we observed cooperative behavior only at the multiple-channel level and in whole-cell recordings, but not at the single-channel level.

Our results show that copper increases the sensitivity of rP2X2a to ATP by increasing the open probability (P_o_), but not channel conductance, through an increase in the cooperative gating between rP2X2a channels. In whole-cell currents, copper increases the EC_50_ without affecting the maximal current. Moreover, the Hill coefficients obtained strongly support cooperative behavior. The EC_50_ values were significantly different only in whole-cell currents and patches containing two to three channels, indicating that copper-induced potentiation is manifested only when more than one channel is present. Furthermore, we found that copper increases the Hill coefficient (nH), suggesting an increase in cooperativity. However, in single-channel recordings, we observed no change in the EC_50_ or nH, indicating no cooperativity, further supporting the requirement for multiple channels to manifest the potentiation. Similarly, Ding and Sachs [22] demonstrated that ATP by itself increases cooperative gating in multiple channel patches; thus, in our experiments, we observed a basal energy of interaction of −0.4 kT, indicating basal cooperativity, but the application of copper increased this cooperativity to −1.4 kT.

This mechanism seems to be similar to that exerted by zinc and copper in rP2X2a receptors and involves the extracellular histidines H120 and H213, which contribute to intersubunit coordination [33] and possibly, as shown in this study, to interchannel coordination of both zinc and copper [14]. By contrast, human P2X2 receptors are inhibited by both zinc and copper [34,35]; the metal inhibitory site (at least for zinc) is a different intersubunit binding site, located near the lateral fenestrations that are the entry point of ions for P2X receptor channels [36].

While changes in conductance (pore dilation) have been reported for P2X4 and P2X7 channels, this is not a common mechanism for the current potentiation of ion channels. In rP2X2 receptor channels, the apparent increase in conductance is only observed when mutations of Cys9, Cys348, and Cys430 are introduced [37], not in wild-type receptors [21,22,32]. Our results agreed with those of Ding, Sachs, and others, and we found no significant change in channel conductance, but we observed an important increase in open probability. Although Virginio et al. suggested that this increase in conductance was the result of pore dilation of rP2X2 receptor channels after prolonged ATP stimulation, we did not observe this effect under our experimental conditions. This discrepancy may be due to our use of only Cs^+^ and Na^+^ as permeant ions and not NMDG^+^, which causes an artifact that resembles pore dilation [37]. Harkat et al. found that activated rP2X2 channels can permeate NMDG^+^ immediately after channel opening, but the rate of NMDG^+^ permeation is 10 times slower than that of Na^+^, challenging the concept of pore dilation [37,38]. Similarly, in our experiment, we did not observe pore dilation, and we did not observe any increase in Na^+^ conductance after the addition of copper (Figure 3 and Figure 4).

The increase in the EC_50_ observed at both whole-cell and multiple-channel levels exhibits an increased Hill coefficient, indicating cooperative behavior, as described for hemoglobin or nAch receptors [25,39]. We quantified this interaction using the one-dimensional Ising model, which allowed us to describe nearest neighbor energetic interaction between ion channels [31]. The interaction between ion channels can be both local and non-local. Local cooperation occurs among identical channels, while non-local cooperation involves different channel proteins. The Ising model assumes a system in equilibrium, with channel states continuously dependent on reactant concentrations.

Binomial distribution is a straightforward method for assessing channel cooperation. However, it requires precise channel number estimation and stable channel states with constant open probability (steady state). In our experiments, we estimated channel numbers by perfusing 300 µM ATP to achieve maximal open probability and then counting the channels. The observed deviation from a probability ratio of 1 suggests non-independent channel behavior. This discrepancy between binomial and experimental probabilities indicates that one channel’s gating influences the open probability of others. Our data, combined with Ising model interaction energy results, demonstrates positive cooperativity, explaining copper’s potentiation of ATP-induced rP2X2a receptor current. Similar behavior has been observed in nicotinic receptors, the bacterial potassium channel KvAP [40], and the ATP-sensitive potassium channel (KATP) [41], suggesting that cooperative gating in ion channels is a fairly common mechanism.

The novel concept of rP2X2a receptors’ cooperative interaction has significant functional implications, given their widespread expression in the central nervous system. It is noteworthy that trace metal-induced receptor channel association might extend to other ionic receptors, as interactions between nicotinic and P2X receptors have been reported [42,43]. Finally, trace metal-induced enhancement of receptor channel cooperativity may play a broader role in neuronal functions, such as long-term potentiation, learning, and memory, or even in pathological conditions, such as Alzheimer’s disease, in which plaques have increased the copper concentration.

We hypothesize that copper modulation of the rP2X2a receptor channel occurs through its coordination with H120 and H213 of at least two channels, a mechanism that could explain the results obtained in the present work. In this context, more experimental and modeling data are needed to explain in detail the mechanism of copper-induced modulation. This novel mechanism could help elucidate some of the unique features of rP2X2a modulation by copper and other trace metals and will help us understand the differences observed between rat and human P2X receptors. Future experiments should explore copper-induced cooperative gating of rP2X2a receptor channels in physiological models (e.g., brain slices and sensory neurons) and pathological models of neurodegenerative diseases, including in vivo studies. In summary, our results indicate that copper increases the ATP-gated current of rP2X2a by increasing its cooperative gating, thus increasing its ATP sensitivity and further inducing potentiation.

## 4. Materials and Methods

### 4.1. Cell Culture

HEK 293T cell lines were obtained from ATCC and cultured in DMEM, high glucose (Invitrogen, Waltham, MA, USA) supplemented with 5% fetal bovine serum (FBS) and 2 mM glutamine. Cells were maintained in a controlled atmosphere of 5% CO_2_ at 37 °C. The full-length rat rP2X2a cDNA (GenBank™ accession number Y09910) was transfected into HEK293T cells using Lipofectamine 2000, following the manufacturer protocol, and then seeded on 12 mm round coverslips and used for electrophysiological experiments after 16 h.

### 4.2. Electrophysiology

For whole-cell recordings and the outside-out patch, the internal solution contained (in mM) 140 NaF, 5 NaCl, 10 HEPES, and 1 EGTA, at pH 7.2 adjusted with NaOH. The extracellular solution contained (in mM): 140 NaCl, 5 KCl, 1 CaCl_2_, 1 MgCl_2_, 10 glucose, and 10 HEPES, at pH 7.4 adjusted with NaOH. Patch clamp experiments were performed using an Axopatch 200B amplifier (Molecular Devices, San Jose, CA, USA). Data were low-pass filtered at 5 kHz and digitized at 20 kHz (Digitdata 1322A) using pClamp 9.2 software (Molecular Devices, San Jose, CA, USA). Patch pipettes were pulled from borosilicate glass (Warner Instruments) using a P97 micropipette puller (Sutter Instruments, Novato, CA, USA) to a resistance of 4–6 MΩ for whole-cell recording and 6–8 MΩ for outside-out patch clamp recordings. All experiments were conducted at 22 ± 2 °C using a holding potential of −60 mV.

For outside-out patches, we first reached the whole-cell configuration, and then we gently pulled the pipette and monitored the drop in capacitance. We considered the patches successful when we observed a decay in capacitance while maintaining access resistance values similar to whole cells (less than 20 MΩ).

### 4.3. Data Analysis

Electrophysiological data were analyzed using Clampfit 10.3 (Molecular Devices), and the statistical analysis was performed using Prism 8 (GraphPad Inc., Boston, MA, USA); statistical significance was determined by ANOVA followed by Dunn’s post hoc test. Data are shown as the mean ± standard deviation. “Single-channel patch” was defined as the absence of current superimposition at a V_h_ = −60 mV with a stimulus of 100 µM ATP, where P_o_ is maximal.

The probability of the channel being open (P_o_) was defined as the ratio of the open channel area (A_o_) to the total area (A_o_ + A_c_) in the all-points amplitude histogram:(1)$$P_{o}=\frac{A_{o}}{A_{c}+A_{o}}$$

For the currents from patches containing multiple channels, P_o,i_, was similarly defined as the ratio of the ith level (A_o,i_) to the total area (A_Total_), which is the summation of closed open peak areas in the histogram:(2)$$P_{o}=\frac{A_{o,i}}{A_{total}}=P_{o}=\frac{A_{o,i}}{A_{c}+\sum_{i=1}^{n}A_{o,i}}$$

### 4.4. Hill Curve Fitting, EC_50_, and Hill Number

Dose-response curves were constructed using 0.1, 1, 3, 10, 30, 100, and 300 µM ATP. The maximal current or P_o_ was plotted as a function of the ATP concentration. The data were fitted using the Hill function:(3)$$y=\frac{{\left[ATP\right]}^{n}}{{EC}_{50}^{n}+{\left[ATP\right]}^{n}}$$
where *n* is the Hill coefficient, [ATP] is the ATP concentration, and *EC*_50_ is the concentration that results in half the maximal response. The Hill coefficient and *EC*_50_ were determined from the slope and intercept of a linearized form of the Hill equation, as follows:(4)$$log\frac{y}{1-y}=n\times log\left[ATP\right]-n\times log{EC}_{50}$$

For the determination of copper effects, we used a fixed concentration of 10 µM CuCl_2_. This was chosen because in our previous studies, we determined that this concentration was optimal to observe copper-induced potentiation, obtaining the maximal potentiation of the ATP-gated currents of the rP2X2a receptor channel [14].

### 4.5. Binomial Distribution Test

If there are N identical channels in a patch, each with a probability P_o_ of being open, and if the channels act independently, the probability of k channels being in the open state simultaneously is given by the binomial distribution:(5)$${P(k)}_{binomial}=(N\;k)P_{o}^{k}{\left(1-P_{o}\right)}^{N-k}$$

P_o_ (P(k)_binomial_) for binomial calculation was estimated from the single-channel concentration-response curve. We compared P(k)_binomial_ with the P(k)_exp_ obtained from the amplitude histograms of the experimental data. A significant deviation of P(k)_exp_ from P(k)_binomial_ suggests either non-independence or non-identical channels. Since binomial analysis has no kinetic information, different kinetic behaviors can produce the same distribution.

### 4.6. Ising Model for Cooperativity

The model can be applied directly to the study of the interaction between ligand-gated ion channel proteins in a biological membrane. We assume that there are m ligand binding sites on the protein, but there are only two distinct states of the protein: state A, the unliganded state, and state B, the fully liganded state. We associate A with a closed state and B with an open state. The Ising model provides a means for considering the contribution of an added source of energy and the interaction energy between the nearest neighbor channels in the overall open channel probability. The positive interaction increases it, while negative cooperativity decreases it; the energy can be calculated at P_o_ = 0.5, where the channel noise is maximum:(6)$$\sigma_{0.5}^{2}=\frac{i^{2}N}{4}e^{\left(\frac{-\epsilon_{BB}}{2}\right)}$$
where ϵ_BB_ is the potential energy of the interaction, i^2^ is the square of the unitary current, and *N* is the number of channels present in the patch. ϵ_BB_ was calculated from:(7)$$nH=e^{-\frac{\epsilon_{BB}}{2}}$$
where *nH* is the Hill coefficient. Any deviation from 0 indicates non-independent gating and changes in the interaction energy; thus, for binomial probability (independent gating), the interaction energy is 0, positive interaction energy means negative cooperativity, and negative energy means positive cooperativity.

## Figures and Tables

**Figure 1 pharmaceuticals-17-01590-f001:**
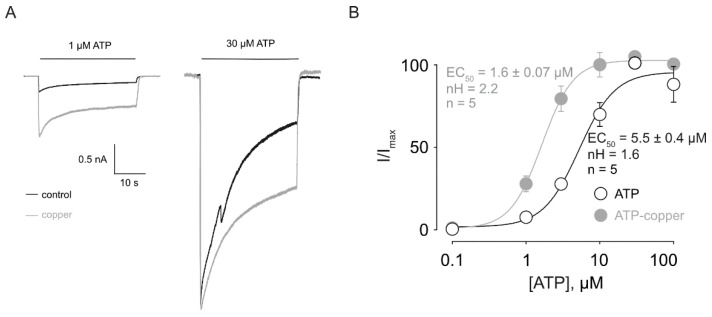
Effect of copper in ATP-induced rP2X2a currents. (**A**) Representative current traces showing the effect of 1 µM and 30 µM ATP (black traces) with or without 10 µM copper (grey traces). (**B**) Concentration-response curve showing the effect of 10 µM copper in the ATP-induced rP2X2a currents. Data were fitted to a Hill equation (methods).

**Figure 2 pharmaceuticals-17-01590-f002:**
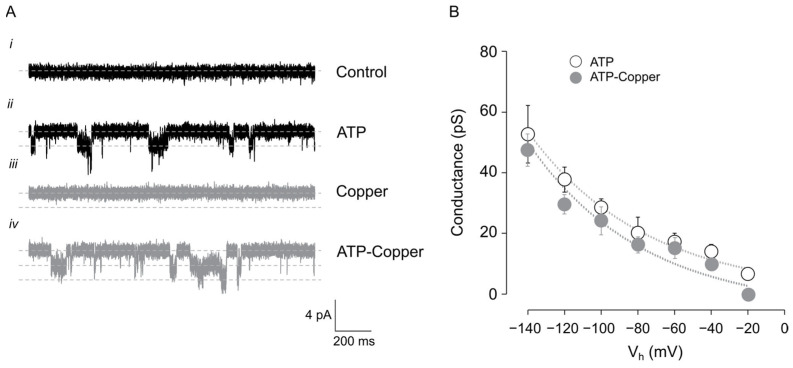
ATP-induced current in rP2X2a containing outside-out patches. (**A**) Current traces recorded at a holding potential of −60 mV in (**i**) current trace from non-expressing HEK293 cell outside-out patches. (**ii**) Effect of 10 µM ATP on HEK293 cells expressing rP2X2a receptor. (**iii**) Effect of copper in rP2X2a expressing outside-out patch. (**iv**) Effect of copper in the outside-out patches containing rP2X2a receptor stimulated with 10 µM copper. (**B**) Summary graph of single channel recordings in patches containing one channel, showing the chord conductance as a function of the holding potential. Data are shown as the mean +/− SD, n = 5.

**Figure 3 pharmaceuticals-17-01590-f003:**
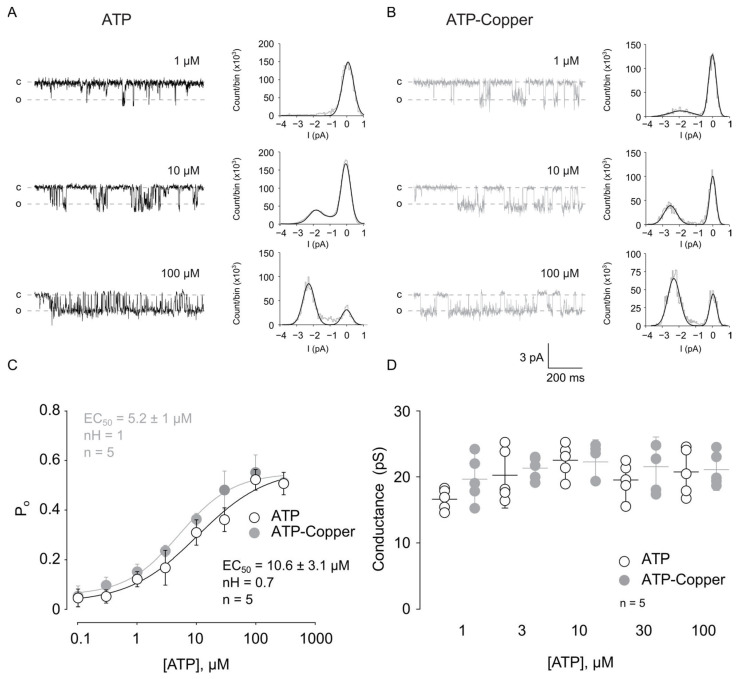
Copper has no effect at the single-channel level. Representative current traces showing the effect of ATP (**A**) and ATP in the presence of 10 µM copper (**B**) at the single-channel current level; right panels show all points histograms, derived from each recording. (**C**) Summary graph of the P_o_ as a function of the ATP concentration; data were fitted to a Hill function. (**D**) Summary plot showing the effect of copper in the conductance at different ATP concentrations. All data are shown as the mean ± SD.

**Figure 4 pharmaceuticals-17-01590-f004:**
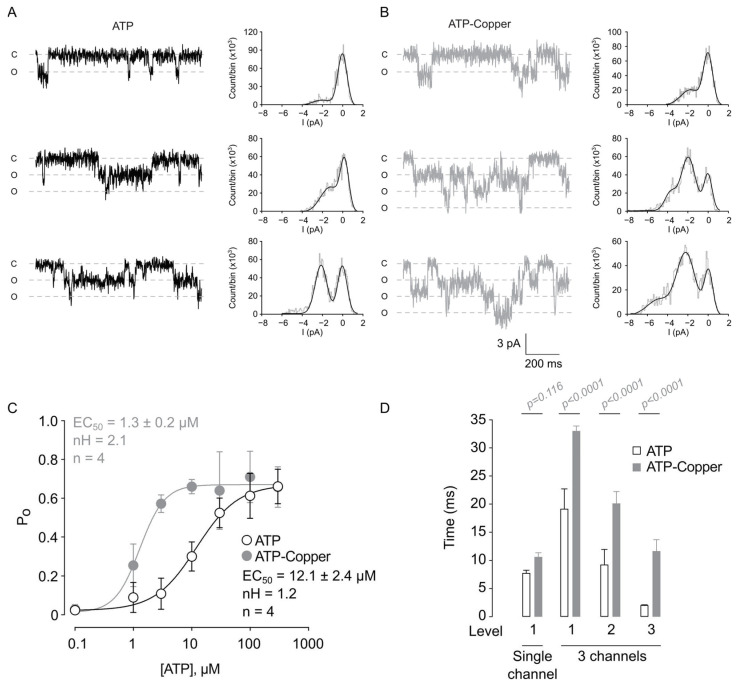
Copper increases rP2X2a ATP sensitivity in outside-out patches containing multiple channels. Representative current traces showing the effect of ATP (**A**) and ATP in the presence of 10 µM copper (**B**) with a multiple-channel current level; right panels show all points histograms, derived from each recording. (**C**) Summary graph of the P_o_ as a function of the ATP concentration; data were fitted to a Hill function. (**D**) Summary graph showing the mean open time of single and multiple channels’ patch current levels. All data are shown as the mean ± SD.

**Figure 5 pharmaceuticals-17-01590-f005:**
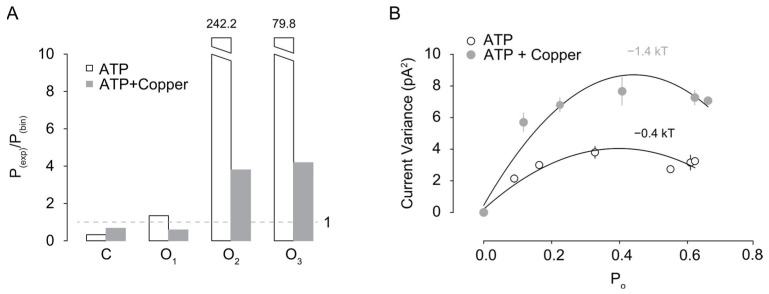
Copper increases the non-independent gating of rP2X2a. (**A**) P_(exp)_/P_(bin)_ ratio of the closed (C) and open state in a patch containing three channels, where each current level is indicated by the subindex (O_1_, O_2,_ and O_3_); in both conditions, ATP and ATP-copper deviate from 1, indicating non-independent gating. (**B**) Summary plot showing the current variance as a function of the P_o_ in a patch containing three channels; data were fitted with a quadratic function, and the interacting energy is shown above each curve. Energy was calculated as described in the methods.

## Data Availability

The original contributions presented in the study are included in the article, further inquiries can be directed to the corresponding authors.

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
