# Peer review of "Copper Increases the Cooperative Gating of Rat P2X2a Receptor Channels"

_pharmaceuticals, 2024, doi:10.3390/ph17121590_

Round 1
Reviewer 1 Report
Comments and Suggestions for Authors
This article by Leiva-Salcedo et al. introduces how copper modulates the cooperative gating behavior of rat P2X2a receptors, with a particular emphasis on its effects on ATP sensitivity and interchannel cooperativity. In general, this is a very interesting topic, but it should undergo some revision. I would recommend the following corrections:
1. Improve the Introduction. Explain why cooperative gating is relevant for P2X2a receptor function.
2. Try to reduce the self-citation to less than 15% or add more new references.
3. Add more details about copper concentrations in Materials and Methods. Also, the methodology for quantifying cooperative gating and ATP sensitivity would benefit from more explanation.
4. Describe the criteria used to determine the presence of cooperative gating in patches with two or three channels versus single-channel recordings.
5. Add a conclusion to make it stronger with future goals, such as in vivo testing or pre-clinical trials.
Author Response
Dear Reviewer,
Thanks for your useful comment. Please find attached the response to your comments.

Reviewer 2 Report
Comments and Suggestions for Authors
Review for manuscript entitled “Copper increases the cooperative gating of rat P2X2a receptor channels”.
This study focuses on the P2X2 receptor channel from rats (rP2X2). P2X2 is an ATP-gated cation channel with multiple physiological roles in neuromodulation, inflammation, and muscle contraction. The authors aimed to elucidate the role of copper as a modulator of the rP2X2 channel. The authors transfected HEK 293T cells with cDNA encoding full length rP2X2 and then conducted a series of electrophysiological measurements on these cells as well as on outside-out patches derived from these cells. The patches either contained one, two or three rP2X2 channels.
A strength of this study is the experimental design/strategy which is presented in a well-organized Result section. The reader first learns that copper increases ATP-induced currents of whole cell preparations. An ATP-dependent dose response curve with and without copper (Figure 1) was analyzed with Hill’s cooperativity model and shows a clear copper stimulated enhancement of rP2X2 induced currents. Next, the authors focused on patches with just one single rP2X2 channel. It was interesting to read that copper had no effect on rP2X2 conductance and also no effect on the ATP-sensitivity of rP2X2 in these patches with single rP2X2 channels. Only patches with multiplerP2X2 channels show copper dependence in a manner that resembled the whole cell preparations (see for example Figure 4C compared to Figure 1B). By conducting this sequence of experiments, the authors present very convincing data for their claim that copper modulates rP2X2 function by cooperative gating. Presenting data on the single channels and then on multiple channels was a great decision made by the authors to make it very clear that multiple channels must be present for copper to have an influence on the cation transport measured in the patch clamp experiments. The authors take their analysis several steps further to investigate the type of cooperative gating. By carefully analyzing the number of open channels and by comparing this number to a theoretical value expected for a non-cooperative case (the authors use a binomial distribution to model the non-cooperative distribution of open and closed channels), the authors were able to establish that copper increases the probability of channels to be open in a positive cooperative model. The probability of open channels was fitted to an Ising model in Figure 5B to assess the interaction strength among channels with and without copper being present. Based on this fit copper increases the interaction strength from 0.4 kT to 1.4 kT. It would be very interesting to find out how copper interacts with the rP2X2 channels on the molecular level and to locate a potential binding site that might be positioned between different rP2X2 units. In their Discussion (line 263) the authors provide a hypothesis in which they propose a copper binding site: “We hypothesize that copper modulation of the rP2X2a receptor channel occurs through its coordination with H120 and H213 of at least two channels.” It would be nice to see a model, but most likely this will be the topic of another publication. The article presented here has a clear focus on the electrophysiological data.
I have one question about Figure 5A: For levels O2 and O3 the ATP-copper condition shown in grey has a lower P(experimental)/P(binomial) ratio than the ATP condition shown in white. Based on the text in the main manuscript I would have expected the opposite legend (higher P(exp)/P(bin) for ATP-copper for all levels than for ATP condition without copper). Could you please explain how to interpret Figure 5A in more detail in your manuscript?
Overall, this is a very strong and interesting publication and I only have a few other minor suggestions as listed below:
Introduction: Grammar correction from singular to plural in line 32 (its should be their) and line 41 (channel should be channels)
Experimental section: Please add a short description on how the outside-out patches were created (or refer to a reference).
Figure 4: Please increase the size of Figures 4C and 4D (please also increase the font size ,especially for the smaller text in Figure 4D).
Figure 5: Please add a brief definition for C, O1, O2, O3 in the figure legend.
Author Response

(The authors gave the same response as above.)

Reviewer 3 Report
Comments and Suggestions for Authors
Excellent work and relevant work demonstrating how copper alters the gating of ion channels. Thanks experimental results are robust and can be clearly framed within the 1-D Ising model. I have no major criticisms except that in the methods section, Ising model, the authors should provide more elaborate information, such as explicit expressions of the half maximal activation concentration and the Hill slope
Author Response

(The authors gave the same response as above.)
